

# Role clarity deficiencies can wreck agile teams

Helena Barke and Lutz Prechelt

Freie Universität Berlin, Berlin, Germany

## ABSTRACT

**Background.** One of the twelve agile principles is to build projects around motivated individuals and trust them to get the job done. Such agile teams must self-organize, but this involves conflict, making self-organization difficult. One area of difficulty is agreeing on everybody's role.

**Background.** What dynamics arise in a self-organizing team from the negotiation of everybody's role?

**Method.** We conceptualize observations from five agile teams (work observations, interviews) by Charmazian Grounded Theory Methodology.

**Results.** We define role as something transient and implicit, not fixed and named. The roles are characterized by the responsibilities and expectations of each team member. Every team member must understand and accept their own roles (*Local role clarity*) and everbody else's roles (*Team-wide role clarity*). Role clarity allows a team to work smoothly and effectively and to develop its members' skills fast. Lack of role clarity creates friction that not only hampers the day-to-day work, but also appears to lead to high employee turnover. Agile coaches are critical to create and maintain `role clarity`.

**Conclusions.** Agile teams should pay close attention to the levels of *Local role clarity* of each member and *Team-wide role clarity* overall, because role clarity deficits are highly detrimental.

# INTRODUCTION

The Agile Manifesto (*Beck et al., 2001*) suggests to *"Build projects around motivated individuals. Give them the environment and support they need, and trust them to get the job done."* So agile teams need to *self-organize*: Where a conventional team would have a manager who assigns tasks to people, defines team work processes, and mediates conflicts that might arise among team members, an agile team needs to do all those things themselves, perhaps supported by a coach who is supposed to catalyze the self-organization processes.

The most widely used agile method, Scrum (*Sutherland & Schwaber, 2016*), which is our sole focus here, is quite explicit about this. It has only three roles: Product Owner, who defines requirements; Scrum Master, the coach; and everybody else, called the "development team". The overall team ("Scrum team") is supposed to self-organize.

Corresponding author
Helena Barke, h.barke@fu-berlin.de

Self-organization is at the very heart of the method; all formalized Scrum method elements are merely tools for making self-organization easier.

There is plenty of evidence that self-organization is difficult (*Barker, 1993*; *Hoda, 2011*; *Moe, Dingsøyr & Dybå, 2008*; *Gren, Torkar & Feldt, 2017*; *Karhatsu et al., 2010*; *Hackman, 1986*). Requiring agile teams to self-organize is one of the most pronounced differences to previous forms of software development. *Hackman (1986)* examined self-organized teams over decades long before agile software development. In those results, being self-organized is not necessarily an advantage; there are low-performing as well as high-performing self-organized teams and an explanation of these differences is lacking. The systematic literature review of *Magpili & Pazos (2018)* looked for such explanations in research from all fields and identified performance-relevant input factors on individual level, team level, and organizational level. Individual roles are one such factor. These roles are not tied to job descriptions and could be implicit. Another factor is the balance of individual and team autonomy.

Although there is a huge amount of literature on agile methods overall (*Dybå & Dingsøyr, 2008*; *Hoda et al., 2017*), few works have considered self-organization as such and as a whole. *Karhatsu et al. (2010)* states that autonomy, communication, and collaboration are the major components for building self-organizing teams. *Moe, Dingsøyr & Dybå (2008)* find the division of labor that results from the specialized skills of the team members to create a major self-organization hurdle. *Hoda (2011)* explains many different phenomena observed in agile teams all in terms of self-organization. She describes *balancing acts* such as specialization versus cross-functionality or continuous learning versus iteration pressure (*Hoda, Noble & Marshall, 2010*). She also describes roles such as Mentor, Translator, or Coordinator which are also not based on job description (*Hoda, Noble & Marshall, 2013*).

In conclusion there is research about self-organized teams in general which shows that roles are one input factor on performance of self-organized teams. There is research about the problems of self-organized teams in software development and there is research about roles in software development. What is missing is how roles are helping or hurting in making self-organized software development teams work.

Since a few years, we have worked with agile teams and analyze how self-organization works or fails to work. In our previous work (*Barke & Prechelt, 2018*) we focused on cross-functional collaboration. Cross-functional teams do have a wide variety of roles (dependent or independent of job description). In this first article, self orientation and team orientation were auxiliary concepts for some facet of the cross-functionality problem. Focusing on these two concepts led to the present work which show how clarifying roles in self-organized software development teams is critical for team surviving.

**Contributions**: We explicate the process by which `Roles`[1] are defined and what leads to `Role clarity`, thus making that process visible, negotiable, and controllable for individuals and teams. We expect our work helps practitioners reach `Role clarity`, allowing smoother day-to-day-work, reduced employee turnover, and accelerated team maturation. The examples shown will hopefully motivate teams to pick the process up.

[1] We use Grounded Theory Methodology (GTM). This markup signifies a concept created during the analysis.

As for the structure of the article, we start by describing our research method ('Method'). The core part of the article presents the definition of role and the two core concepts in detail along with the evidence that led to them ('Results'), followed by discussion of the findings, the related work and the application for practice ('Discussion'). After a short discussion of limitations ('Limitations'), we formulate the take-home message ('Conclusion').

## METHOD

### Method choice

Our research originally started from an interest in diversity and how teams made use of it (or not). We had no particular preconception what types of data or what kinds of phenomena would be most helpful to find this out, so we selected the most open and flexible of the qualitative research methods, Grounded Theory Methodology (GTM). GTM is applicable to research interpersonal processes in software development teams and is now established in software engineering (*Stol, Ralph & Fitzgerald, 2016*; *Badreddin, 2013*). It is highly suitable for examining agile contexts (*Hoda, Noble & Marshall, 2012*; *Javdani Gandomani et al., 2013*) as GTM and agile methods have similarities: they are iterative approaches, minimize unreliable long-term planing, and put people and their actions at the center (*Hoda, Noble & Marshall, 2012*).

We use the Constructing Grounded Theory variant of *Charmaz (2014)* as we share her view of a reality that is constructed by participants and researchers (see 'Use of GTM Practices').

### Data collection

We used `Theoretical Sampling` (*Charmaz, 2014*, p. 195): Gathering the data iteratively, driven by the needs of the ongoing analysis. The teams were chosen to cover a variety of different company structures and software domains and had different levels of maturity. As far as possible, team members to be interviewed were also chosen iteratively such as to obtain complementary perspectives on the team. Although our data set is small, the diversity across teams and team contexts is high. Our teams are very different at the surface which requires a thorough analysis so as to reveal recurring structures across teams. Longitudinal data was gathered along longer-term team processes in four of the five teams. MAXQDA (www.maxqda.com) was used as software support.

A personal network of the first author from the mentoring program for students of the HTW University of Applied Sciences Berlin helped to get access to a wide range of companies. The personal contact via the mentoring program made it easy to establish research cooperation and industry mentors from that program were members of two of the observed teams (but to avoid biases were never chosen for interviews).

Most of the examples below come from five software development teams which we refer to as `t1` to `t5`.

All data collection was done by the first author. For an overview please see Table 1.[2]

This study is based on 16 qualitative interviews and nearly 20 h of audio recordings. About 14 h are individual interviews, which were nearly completely transcribed. The remaining recordings were made during feedback meetings and discussions and were used

[2]The data used in the present study is the same as used in *Barke & Prechelt (2018)* plus a few extensions.

**Table 1  Data collection in Scrum teams.**

| Team | Interviews | Timespan | Review | Retro | Planning | Refinement | Stand-up | Additionally |
|---|---|---|---|---|---|---|---|---|
| t1 | t1dev1 t1dev2 | one day in 2015 | | | | | 1 | |
| t2 | t2dev1 t2test1 t2test3 | Nov 16–Oct 17 | | | 5 | 1 | 2 | team analyzing discussion, anonymous online questionnaire, Retro moderation |
| t3 | t3test1 t3sm | Feb–May 17 | | 2 | 3 | 1 | 1 | feedback June17 |
| t4 | t4sm2 | June–Sep 17 | | | 1 | 1 | 2 | group discussion |
| t5(t6) | t5sm1 t5dev1 | Sep 17–Jun 18 | 4 | 3 | 4 | 2 | | feedback with SMs January and June18, feedback complete team Jan18 |
| | | Total amount of meetings | | | | | 33 | |

to further differentiate some concepts. In addition, more than 100 pages of hand-written notes reflect observations from team meetings where participants did not like to be audio-recorded. These hand-written notes contain as much near-verbatim language as possible to gain unbiased raw data even without audio recording it.

The longest research collaboration with one team (`t2`) lasted 11 months. All teams stated they work with Scrum, which was the only property they all had in common. Other than that, they represent a wide spectrum as shown in Table 2. The research work received an ethics approval waiver from Zentraler Ethikausschuss (ZEA) of Freie Universität Berlin.

**Interviews**: The interviews were face-to-face, hour-long, semi-structured interviews, focused on individual action and perceptions. Except at `t3`, at least two team members were interviewed per team to get different perspectives. It was often difficult to keep interviewees in that personal perspective; they had a strong tendency to switch to a team perspective. The interview questions varied between interviews because of *Theoretical Sampling* (see 'Use of GTM Practices' and Fig. 1). The basic interview guideline is attached to this article. Our quotes will be tied to individual team members. For instance, `t4dev2` means team `t4`, second person which is an developer. We use `dev` (developer), `po` (product owner), `sm` (scrum master/coach), `test` (tester).[3]

[3]This generic naming was developed during research. It is and will be consistent in all our publications (*Barke & Prechelt, 2018*)

**Observations**: In three teams (`t2`, `t3`, and `t5`), we observed at least two successive sprint changes. The sprint changes included Review, Retrospective, and Planning. Where possible, we also observed Refinements and Daily Scrums. `t6` had planning and review meetings together with `t5`, as they worked on the same project. `t6` had only two to three team members during our observation phase and was in a transformation process, so it was not investigated further. The observations were not limited to team meetings. Usually before and after the meetings, time was spent with the team or individual team members. Especially during sprint changes, complete days were observed including other work situations and even lunch break.

**Member check and feedback**: After completing the first round of data collection in these teams and initial analysis of the status quo we did a member check: We shared our preliminary results with the respective teams and asked for feedback in different ways as

| Team | Size | Company and work context | Agile background |
|------|------|--------------------------|------------------|
| t1 | 90 employees | business management software | since 6 years agile |
| t2 | 1500 employees, distributed over 3 countries | hardware planning tool for internal use | since 1 year agile, long waterfall background |
| t3 | 15.000 employees, 15 Scrum teams | public instituation with external members of a service company building a system of complex business logic | since 2 years agile, long waterfall background |
| t4 | 25 employees | start-up developing web portal | since 4 years agile, agile from the beginning |
| t5 | 150 employees | sub-company of a much bigger one, e-commerce, importing data from suppliers databases | since 2 years agile, agile from the beginning |

**Table 2  Background of the Scrum teams.**

shown in column 'Additionally' in Table 1. Notes from the feedback sessions were fed back into the research process to enrich and validate the identified concepts.

**Expert interviews (outside teams)**: We obtained indirect insights on many additional teams (from a range of companies of different sizes, domains, and agile backgrounds) via six expert interviews with agile coaches and consultants from outside the context of the above teams. For these interviews a slightly different interview guide was used. To avoid sweeping generalizations, we asked the experts to report on *concrete* teams and describe *specific* situations.

## Use of GTM Practices

Charmazian GTM suggests various practices, in particular *Initial Coding*, *Fosused Coding*, *Constant Comparison*, and *Theoretical Sampling*. We now explain their interplay by means of Fig. 1, which shows the practices (as rectangles) and the resulting concepts obtained with them (in red).

*Initial Coding* (*Charmaz, 2014*, p. 109) and *Focused Coding* (*Charmaz, 2014*, p. 138) were performed iteratively over all data several times. Charmaz' questions and advice were used to support open coding (*Charmaz, 2014*, pp. 116,120,125,127). Despite Charmaz' and *Hoda, Noble & Marshall (2012)*'s reservation about word-by-word coding, we found it helpful as a starting point; it helped build an understanding of the GTM method (as opposed to just counting topics). Comparing *incident by incident* helped to find difference in the data and extracting the common process. Care was taken to *assign actions instead of assigning types to people* (*Charmaz, 2014*, pp. 15,117), coding actions with gerunds. During initial coding manifold quotes and incidents were found and examined in detail and lead to many different concepts. (A few of the quotes can be found as examples in the results 'Results'). The codings we concentrate on in this paper were mostly around `Self-Reflection`, a wide variety of different `Actions supporting the team`, and different situations and ways of giving `Feedback`. By this detailed examination, the data were "broken up" as a preparation for structuring them during Focused Coding.

During *Focused Coding*, theorizing *Memos* were written with exemplary quotes. Memo writing (*Charmaz, 2014*, p. 162) was also used to document the process and gain ideas and theoretical understanding. The findings from this work were discussed in different contexts: in research meetings with other academic researchers; in agile-interest meetups

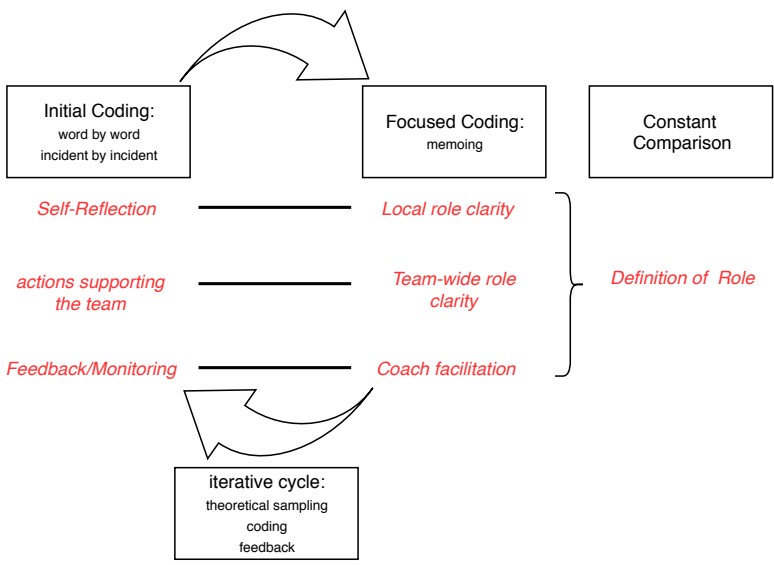

**Figure 1** **Coding Process.**

with unrelated Scrum coaches; in feedback meetings and retrospectives with members of the investigated teams. Ideas from those discussions improved our _Theoretical Sensitivity_ and were subsequently tested in the analysis (iterative cycle).

In this paper we restrict ourselves to a small selection of concepts that help to tell and understand the overall narrative. In the text, names of concepts are given as `Concept Name`. Such concepts are always based on extensive analysis and grounded in data. To make the latter palpable, at least one quote is provided as an example for each concept. Such quotes are given like this:

No. 1: "_I said this!_"(`speaker`).

Each Quote has a number to refer to it. Most quotes were originally in German; since a close translation of spontaneous spoken language is hardly helpful, we have polished grammar and phrasing towards written language during translation. We guaranteed anonymity to the interviewees and teams such that quotes can not be traced back to them, so we cannot give public access to all of our data.

Our discussion is structured around grounded and generalized phenomena, not around specific people or teams as a case study would do it.

## RESULTS

### Overview

We present two important results from our research, which we illuminate by examples in the following Sections:

- A definition of `Role`
- and a process how to `Clarify Roles` on two levels.

A definition of role evolved during research and considers the peculiar circumstances of self-organized software development teams, in particular the transfer of autonomy from management to team level. We define a `Role` as an `Area of responsibility` plus the `Expertise` required to cope with it. Areas of responsibility talk about both what to achieve (goals) and how to pursue it (work style, culture/conventions, etc.). Complying with a `Role` requires to

- `Accept the responsibility` (`R1-Accept`),
- `Possess expertise` (`R2-Expertise`), and
- `Act accordingly` (`R3-Acting`)

so that the expectations regarding the role are fulfilled—one's own expectations as well as the team's. To refer to these attributes of roles we use the short labels `R1-R3`. This definition of `Role` covers a continuum of `Roles`, which mirrors complex team dynamics: For example, one can feel responsible (`R1-Accept`) for technical factors such as code quality, architecture, or testing or for team and social factors such as motivation or for a combination of several factors. The definition goes beyond enumerated explicit roles (such as the standard Scrum Roles) and allows for implicit, intertwined, or dynamically changing roles, among others. In fact, all roles we found in our teams were partly or fully implicit and can be characterized as intertwined and potentially dynamically changing.

For the present article, our insights cluster around two concepts:

- `Local role clarity` is an individual's proper understanding of their own role in the team.
- `Team-wide role clarity` requires role clarity of all team members plus the *mutual* understanding and acceptance of each other's roles. We use the shorter term `Role clarity` when the separation of individual and team level is not of interest.

Our results characterize steps needed to achieve `Local role clarity` and `Team-wide role clarity` and also how teams can fail to achieve them. The narrative results points out that `Role clarity` is difficult, because it requires two steps, each having multiple substeps, and any missing substep tends to make clarity unachievable.

We refer to the following process when we discuss the team episodes we use for grounding and illustration (see also Fig. 2):

`Self Reflection` (to gain `Local role clarity`):

- Team member M is conscious about his/her `Role` (`self-conscious`) and
- accepts the `Role` (`self-accept`).

`Team Reflection` (to gain `Team-wide role clarity`):

- Each team member is aware about M's `Role` (`team-aware`).
- accepts this role as appropriate (`team-accept`), and
- knows how to support M in it (`team-support`).

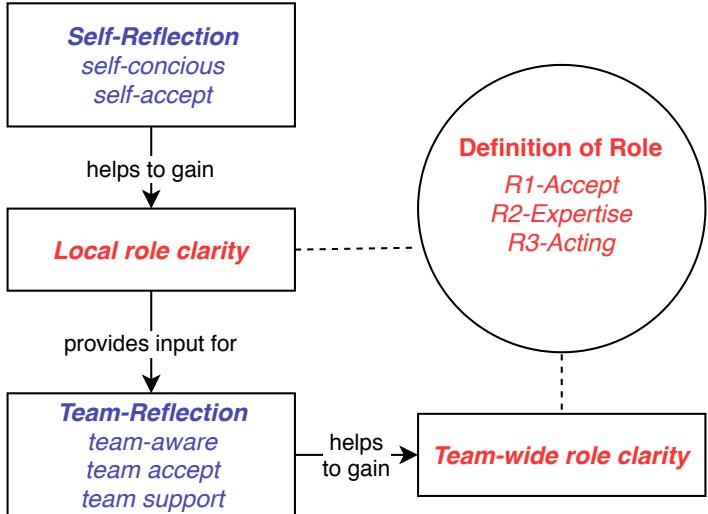

**Figure 2** **Key concepts.**

Although all our observed teams (or at least some members in them) appear to have had a vague notion that something like a team-wide understanding of roles was important for self-organization, none of them had previously recognized `Self-Reflection` and `Team-Reflection` (let alone their substeps) and so none had a conscious process for achieving `Team-wide role clarity`.

### `Self-Reflection` **to gain** `Local role clarity`

The process of gaining `Role clarity` begins at individual level by gaining `Local role clarity`. We start with a statement of frustration from `t2test1`, who had formerly acted as the Scrum Master of `t2`:

> No. 2: *"I feel like a babysitter. [...] we had just understood our roles, then he (`t2test3`) came into the team. We do not know his role, he does not know his role. I encourage him to talk, but he has a different personality: I try to find my place by myself; he wants somebody to do it for him."* (`t2test1`)

The statement points out the need for self-reflection as the first step (`Self-Reflection`). Nobody else can do this for me in a self-organizing team. It also shows that role consciousness (`self-conscious`) is not easy for everybody. The frustration underlying the statement, only part of which is visible in the quote, shows that a lack of role understanding makes self-organization difficult. This quote also contains an implicit expectation that a Scrum Master should help finding one's role and could fail with it.

`Team t2` came from a waterfall background in a conservative company, had switched to agile only a year ago, and `t2test3` was not the only member having a hard time becoming `Self-Reflecting`. `t2dev1` was much less passive and more willing trying new things, but was struggling, too:

No. 3: *"Meditation. This helps. It helps to think about it: What bothers you now? Why are you so pissed off now? [...] But of course that's a long process: This is a strong change in thinking, a mental change; it is incredibly difficult, because you have to have a lot of self-reflection."* (`t2dev1`)

When the team first started with agile, this developer took a sabbatical of several weeks to cope with the situation of no longer being the sole governor of his technical domain. In contrast to `t2test3` from quote No. 2, he managed the first step of `Self-Reflection` and became conscious about his new role (`self-concious`). But he is struggling with the next step: accepting his (new) role (`self-accept`).

To summarize these two examples: In order to function as a participant in self-organization each team member needs to reflect about her/his role (`Self-Reflection`) to gain `Local role clarity`. `Self-Reflection` contains two sub-steps. I first need to be aware of my role (`self-conscious`) in order to be able to accept it (`self-accept`).

Lack of `Local role clarity` may also lead to conflict. The above `t2dev1` had one with `t2dev2`, which was successfully mediated by a coach, in this case `t2po`. The conflict was so strong that it came up in several interviews and in informal conversation with various team members. `t2dev1` summarizes it like this:

No. 4: *"I told him (`t2dev2`) clearly [that I don't like that] he often does not involve people. He does things without telling anybody: We talk about a user story in the Grooming and he says "I completed that long ago. Only need to commit it." Excuse me? How about involving us in that decision?? You force this upon us –that's not so cool. We as a team have to take responsibility for it. [...] After that, we were not on speaking terms for a long time. Eventually,* `t2po` *mediated between us and his attitude changed towards how one should work –as the team expects of him."* (`t2dev1`)

As `t2dev1` recalls it, at the heart of the mediation was undiplomatic feedback, roughly like this:

No. 5: *"`t2dev2`, I don't respect what you do, because it is often not well thought-through. It is imprecise, unclean, and badly done. It has not been agreed upon and creates a lot of additional work later. [...] Therefore, I cannot respect you as a developer."* (`t2dev1`)

The significance of this dialogue lies in the fact that the situation required a facilitation initiative of `t2po` to make this feedback happen. But then the resulting change in role acceptance was comprehensive:

No. 6: *"He no longer makes lonely-cowboy decisions. He tries to participate in discussions on topics he does not know much about and which he finds hardly interesting. To listen, maybe even say something. He presents his ideas before implementing them. And he…how should I put it? OK, arrogantly: He listens to me.* (`t2dev1`)

This conflict is about an issue that is highly important for an agile team: accepting responsibility (`R1-Accept`). The conflict persists until `t2dev2` accepts his role more fully (`self-accept`). Afterwards, there is evidence of better team cooperation (Quote No. 6).

`t2dev1` is the most senior developer on the team and has been developing that particular software since 30 years. The episode shows that the lacking `Role clarity` effect can be strong and could easily be mistaken for a lack of technical expertise. But `t2dev2` is not a junior developer at all; the problem is a `Local role clarity` issue.

**Roles in self view vs. team view to gain `Team-wide role clarity`**

Above, we have seen examples how `R1-Accept` can be a problem. By obtaining `Local role clarity`, responsibility of one's `Role` is accepted (`R1-Accept`). In order to become clear about one's expertise (`R2-Expertise`) and act accordingly (`R3-Acting`), `Team-reflection` is needed as next step. Starting with a quote about `t6dev` from the joint Scrum Master of `t5` and `t6`:

> No. 7: *"For him, self-image and how others perceive him are a bit different. When I give him feedback that something was not good, he is surprised: "I thought I am the super-high performer." I told him to ask others as well and adjust his self-image or his performance. And indeed he tries to improve –but it does not help: The next time around he is surprised again."* (`t5sm1, paraphrased`)

In `t6dev2`'s self-view his expertise (`R2-Expertise`) is high. He would consider himself as having `Local role clarity`, but from his team's point of view, the `Possess expertise` (`R2-Expertise`) component is lacking; he has expertise only for a different, weaker role. As a consequence, `Team-wide role clarity` is not achieved—which shows again: Even with coach facilitation, complete role understanding may remain difficult for a team. Some time later, the company decided to let `t6dev2` off, whereas another problematic developer, `t6dev1` could stay, although his technical competence was lower than `t6dev2`'s; he was also a slow learner, but he was good at `Self-Reflection` (He says about himself No. 8: ""*It's hard to learn about new tricks.*"" (`t5sm1`)) and so the team saw a better overall perspective for him.

This shows how `R2-Expertise` and `R3-Acting` can be a problem as well –not because of a *lack* of technical competence (`R2-Expertise`), but as a hurdle for `Team-wide role clarity`: Successful negotiation of `Roles` enables a team to do its job with whatever level of technical competence is available at the time, but forming joint expectations about `R3-Acting` may involve conflict, which is created for instance from perception differences or preference differences.

The first two substeps of `Team-reflection` towards `Team-wide role clarity` mirror the substeps of `Self-Reflection`: role awareness (`team-aware`) and role acceptance (`team-accept`), only this time for someone else's role. To make the `Self-Reflection` match the `Team-reflection`, all team members must have approximately the same view of what the level of expertise (`R2-Expertise`) involved in the unique role of a team member needs to be. This becomes a problem in particular when the team member has a distorted self-image as in the case above.

The episode impressively underlines the difficulty of achieving sufficent understanding and acceptance of members' `Roles` and how such difficulties undermine a team's self-organization.

### Effects of having `Role clarity`

Having `Role clarity` has several positive effects for the team and each team member.

#### *Handling of technical expertise (*`R2`*)*

When `Role clarity` is achieved, technical expertise becomes less important (`R2-Expertise`), because expectations and reality align (`R3-Acting`) like in this example:

> No. 9: *"He is sometimes a bit slow, but, hey!, he does his job. Everything takes a little longer, but that is not the issue, as long as the code works in the end. I don't care how long it takes."* (`t2dev1`)

Also evaluating and growing expertise (`R2-Expertise`) becomes more matched to the needs of the `Role`. Here is a positive example:

`t3dev3` is a new and junior team member, with the additional difficulty that his German skills (German is the team language) are only intermediate. When people speak fast or use difficult terms he often cannot follow. `t3dev3` is aware of his new and junior status as well as his limitations with German (`self-conscious`) and talks about this openly (a consequence of `self-accept`). The team understands this as well (`team-aware`) and is willing to help (a consequence of `team-accept`). In total `Self-Reflection` and `Team-reflection` are aligned and `Role clarity` is gained. This led to tailor-made help for `t3dev3` to grow `R2-Expertise` as follows. In a team meeting, the team considers many possibilities and their constraints: (1) `t3dev3` has stated that he wants to contribute to the team's output, so he wants to work on complete tasks. (2) The next sprint will start new things in green-field fashion so other team members will be learners as well. (3) The team could plan fewer tickets to allow for pair programming, so `t3dev3` could learn a lot and the language barrier could be lowered. (4) However, `t3dev3` likes to work out an overview and understanding by himself, so this idea is postponed. (5) `t3dev3` has sometimes stopped asking questions when they were not answered on first try. (6) The team encourages `t3dev3` to perhaps submit questions in writing and (7) also ask *more* questions, because more things are unclear for more team members in the future anyway.

We do not know how the situation played out eventually, but it is obvious from these observations that without the team's concerted attempt at understanding how to support `t3dev3` in his `Role`, it would have been more difficult to make him productive; this corroborates that self-organization becomes difficult when `Team-wide role clarity` is low.

#### *Robustness Against Dysfunctional Behavior*

The following episode illustrates an interesting aspect of `Team-wide role clarity`: Increasing `Team-wide role clarity` can make a team more robust and resilient against dysfunctional or even destructive behavior of a team member.

An example: In an act of `Self-Reflection`, one day `t4dev3` decides it is time for him to move on in his career: He wants the role of a team lead and assign tasks to others. He speaks with management, who denies him such a role, as it would be alien to the agile work style. As a consequence, `t4dev3` starts disrupting meetings or staying away from them.

His team colleagues are confused, because they do not know the reasons for his dysfunctional behavior. The coach `t4sm2` makes this behavior a topic of the next retrospective. `t4dev3` explains that it is a conscious tactic to make management take him out of the agile context and give him a team lead role. The coach reports:

No. 10: *"We talked openly about it. Everybody had felt bad with the situation; they almost did no longer want to go to work in the mornings. But talking helped. It did not resolve the situation or change* `t4dev3`*'s behavior, but the others no longer took it personally and could cope with it with more confidently.* (`t4sm2, paraphrased`)

In the terms of our analysis, `t4dev3` had `Local role clarity`: he dropped the acceptance of his previous roles (`anti-self-accept`) and assigned himself a disruptor role (`self-conscious, self-accept`). But there was zero `Team-wide role clarity` in both respects. The retrospective changed this: The team recognized the disruptor role and although they did not accept it (`non-team-accept`), the partial clarity allowed them to work better as a team. Eventually, management suggested to `t4dev3` to go separate ways, which they then agreed to do.

Although the situation is fundamentally negative and although `Team-wide role clarity` remains incomplete, a lot of progress is made, which emphasizes the usefulness of role understanding. Also, once again a coach triggers the resolution of a difficult situation. Finally, the episode shows that perfect role acceptance may will be impossible.

**Effects of not having `Role clarity`: Wrecking teams**

After the previous two positive examples, a highly negative (if extreme) example shows how not having role clarity hampers self-organization. We have already heard about `t2test3` in our very first example (Quote No. 2), the person who wanted others to devise his roles for him. At the team level, the effect was profound: No team member had even a rough idea of his role.

One of them summarizes this bitingly:

No. 11: *"During Planing, we had a test user story.* `t2test3` *says "I want to do it in this-and-that way.". OK. Do it. I don't care. What does it have to do with us? Nothing. Why are you even here [at the planning]? I've no idea! Why are you in our sprint at all? I do not know.* (`t2dev1`)

Note that `t2test3` *did* have technical capabilities. But he never asked the questions needed to make his work useful for the team:

No. 12: *"He does not play a part. In a Scrum team, everybody must play a part –but he doesn't. He just sits there. He's present. But he is not present so unobtrusively that I could ignore him.* (`t2dev1`)

And as if the loss of one team member's contributions was not enough, the latter aspect even appeared to drag down the team additionally, in particular in planning activities:

No. 13: *"that sometimes hampers me. [...] I often have the impression that things are discussed differently because he cannot contribute anything* (`t2dev1`)
<sup>4</sup>including in teams t2 and t3 that generally had long member tenures.

Several team members, when we asked them to shortly characterize each of their colleagues in the team, failed to mention `t2test3` at all!

The fate of `t2test3` was the same as the one of the disrupting team member `t4dev3` or the one who believes he is a high-performer (`t6dev2`): They all were asked to leave the company even though they were technically skilled. These were not the only cases of people from our teams leaving their company. We saw five cases overall.[4] With only one exception, roles issues were involved and played a major part in the event: Either a team member lacked `Self-Reflection`, or a member performed `Self-Reflection`, but did not communicate their expectations to the team, or the team was unable to create consensus about a role (`Team-reflection`).

We conclude that a lack of `Team-wide role clarity` frequently leads to loss of team members, which may sometimes be helpful for the team, but more often will be a major concern for the teams and companies involved, because capable software development people are scarce.

### The effects of emotions

As we have already seen in the examples, strong emotions are often involved in `Role` issues: (E1) Team `t4` reported strong negative feelings due to the disruption, (E2) team `t3` carefully and explicitly handled `t3dev3`'s emotions in their role coordination, and (E3) the negative emotional subtext in the extremely-unclear-role case of `t2` is difficult to overlook.

We even encountered one episode that had the emotional aspect at its very center: (E4) `t5dev1` is very unhappy with some aspects of how `t5dev2` acts in his role. After some time, he took the courage to unannouncedly visit `t5dev2` at home in the evening to invite him to a beer and No. 14: "*confess*" (`t5dev1`) his feelings. When `t5dev1` confronted `t5dev2` with his troubling work behavior, he was not surprised of the behavior itself at all; he was aware that he acted in this manner. But he was quite surprised by `t5dev1`'s negative feelings about it. So `t5dev2` had `Self-Reflection`, but for `t5dev1` it was an emotional hurdle to trigger `Team-reflection` and "confess" his feelings.

In the longer term, the dynamics of this role conflict were explosive: `t5dev1` does not follow up the resolution of the conflict and falls silent again until eventually he has an emotional outburst and leaves the team and even the company. But `t5dev1` (just like `t5dev2`) was a key technical person in that team; over the course of a few months, the whole team dissolves and all members except one also leave the company.

Negative emotions can bind a lot of energy and as examples (E1) to (E4) show, it is not unusual for this to occur due to `Role` conflicts. As (E1) and (E2) show, proper negotiation of these conflicts helps release this energy and make it available for the team's work. But as (E4) shows, improper negotiation not merely hampers a team's effectiveness, it endangers the very existence of the team.

## DISCUSSION

Here, we summarize our findings, discuss how they relate to earlier research, and give some ideas how they might be used to improve software practice.

## Summary of findings

Recognizing the concepts and analyzing their implications has led us to the following findings:

1. Self-organization becomes very hard when there is incomplete understanding of the roles in the team;
2. this happens suprisingly often;
3. coach-facilitated reflection on roles can presumably help;
4. achieving sufficiently complete understanding and acceptance is difficult even then.
5. Once achieved, good role understanding and acceptance helps a team to be effective and work together well.
6. The process of gaining `Role clarity` is difficult and can be emotionally challenging.
7. Technical skills can be less important than being self-reflective.

As a positive example, one coach characterized a former team like this:

No. 15: *"They had found each other as a team. They could rely on each other. It was clear who could do what. One writes plenty of code, another thinks ahead, another is good at diplomacy and negotiating with the product owner, and yet another is more like "I'll run and get us person so-and-so.". Without speaking, they knew who had which skills and who would pick up which things to be done. No need for much coordination, it arose dynamically just so. Five strong personalities. Initially, they had had a very strong storming phase, real quarrels about how they would work. But then: fantastic productivity and fun at work."* (`t4sm2`, `paraphrased`)

This quote No. 15 summarizes the results perfectly. The team members found their different responsibilities respective roles (independent of job description) which was in line with their skills and the team expectations. This was not gained without conflict and built by "strong personalities".

The present work shows that (and how) role acceptance (`R1-Accept`) and expectation differences (`R3-Acting`) can create major difficulty for collaboration, much more so than actual expertise issues (`R2-Expertise`, see also quotes No. 7 and 8 and 'Handling of technical expertise (R2)'). The specific content or level of expertise (`R2-Expertise`) is much less relevant. Instead a team member's *assessment* of their own or a colleague's expertise can lead to different expectations and those differences create much of the dynamics. Once a team manages to find `Roles` that are appropriate for the team members at hand and goals at hand, a part of the self-organization puzzle will be solved (see 'Effects of having `Role clarity`' to 'The effects of emotions').

## Related work

Here we discuss which parts of our results pertain to related work and how they fit with it, extend it, or go against it.

There has been interest in the relevance of personality type (such as MBTI type or Big5 profile) for team work, e.g., (*Lenberg, Feldt & Wallgren, 2015*; *Cruz et al., 2011*; *Choi, Deek & Im, 2008*; *Salleh et al., 2009*). This is not a perspective we share and does not fit our definition of `Role`. Our research method requires to focus on *concrete* action, rather

than assigning types (which represent mere *preferences*) to people (*Charmaz, 2014*, p. 15). Our Roles are team-wide agreements, much like team norms (*Flynn & Chatman, 2003*; *Feldman, 1984*), but in contrast to those each one is created for a specific team member. *Lenberg & Feldt (2018)* conclude in a quantitative study that there is a strong business case for achieving team norm clarity and we report qualitatively related evidence for Team-wide role clarity.

*Hackman (1986)* who researched self-organized (mostly non-software) teams for decades, states that (1) *people taking personal responsibility* is one behavioral sign of self-management (akin to R1-Accept), (2) team members need to be equipped with knowledge and skill relevant for the task (akin to R2-Expertise), and (3) team members employ task performing strategies (akin to R3-Acting). GTM results emerge from the data, so we found *Hackman (1986)* only after our analysis. His results and ours nicely corroborate each other.

*Whitworth & Biddle (2007)* concentrate on the positive effects of agile teamwork from a system theory point of view. They found a positive effect of *"Team Awareness and Acceptance"* (one aspect of Team-wide role clarity) which was associated with an *"increased sense of individual responsibility and self worth"* (one aspect of Local role clarity). Even though their research concentrates on positive effects, they found some pitfalls of agile methods. In particular, as in our 'Self-Reflection to gain Local role clarity', 'Robustness Against Dysfunctional Behavior' and 'Effects of not having Role clarity: Wrecking teams', team members feel stressed and exhausted when they are unable to integrate certain individuals into the team. It seems likely that Role clarity contributes to *psychological safety* (*Newman, Donohue & Eva, 2017*) which is helping teams as well.

*Langfred (2000)* discusses individual and group autonomy as a paradox in self-organized teams. *Moe, Dingsøyr & Dybå (2008)* recognize this paradox in agile software development teams as well: The highly specialized skills in software teams that lead to a high level of individual autonomy are a hurdle in becoming self-organized and sharing autonomy at the team level. Our results show how role negotiation is important and helps the teams to balance their members' autonomy and find agreements (Team-wide role clarity).

*Hoda, Noble & Marshall (2013)* describe emergent roles in agile teams such as Mentor, Coordinator, Translator, Champion, Promoter, and Terminator. In contrast, our work is not interested in such *generic* roles, rather in the process how teams tailor and customize roles to their members. We share Hoda's understanding of "informal, implicit, transient, and spontaneous roles" and her roles fit our definition of Roles in many respects, except that they are not tailored to a single individual and can hence be named.

In a nutshell: The shift to more individual and team autonomy in agile methods causes problems, in particular at the team level. Our research closes a gap between previous research that gave examples of helpful roles on the one hand and general research on the autonomy problem on the other by characterizing the roles-finding problem as a whole.

## Implications for practice

Agile coaches are usually considered facilitators for a team's self-organization and indeed this is what we find. They play a key role in making `Team-reflection` happen and achieving `Team-wide role clarity`—or not. Many of the generic methods from coaches' toolboxes are applicable. *Andriyani, Hoda & Amor (2017)* shows how different levels of reflection can be integrated in retrospectives and *Babb, Hoda & Norbjerg (2014)* discusses where *Reflection in action* and *Reflection on action* have their place in agile work. These results and ours suggest that team members learn about each other's skills and expertise during collective estimation and planning. This makes planning a crucial moment in the roles-finding process (and many of our episodes illustrate this), so paying attention to this aspect of planning will likely pay off highly.

Some team members fail at (or do not even attempt) `Self-Reflection` and then coaches need to work with them individually. More often, however the reflection difficulty occurs at the team level and coaches can then use natural situations for their interventions, such as planning, story refinement steps, and of course retrospectives. In any case, `Team reflection` about roles appears to be a near-continuous need.

When we discussed our results with agile coaches and non-coaches from outside our teams, they agreed that making the process explicit ought to be helpful for practitioners, because they can then replace intuitive action with rational action and explicit discussion. The coaches recommended the following tools for clarifying roles:

**Role card game** enforces the reflection of upsides and downsides of different roles and behaviours.[5]

**De Bonos 6 hats** helps to discover and gain empathy for different work styles and realizing their benefit.[6]

**Lego serious play** Each team member builds a metaphorical scene about the team and explains it: Who is standing where? Who is doing what? Why? What is she doing herself? The rest of the team can ask questions. It helps to talk about the scene in the third person to reach openness and discover team issues.

**Job title game** helps individual team members formulate salient aspects of their role and check how far the team's view is aligned. [7]

Of these, *De Bonos 6 hats* focusses on the `team-accept` aspect, the others address the role-definition problem holistically.

## LIMITATIONS

**Completeness:** We have not reached *Theoretical Saturation* (*Charmaz, 2014*, pp. 214,345), so while our work illustrates dynamics that are sure to exist (due to grounding), the explanation of those dynamics may be incomplete; it is not a full theory.

**Generalizability**: All our teams were Scrum teams, making generalization to other styles of agile development less reliable (although we are optimistic in that respect). All our teams were German teams. Further studies need to check where the results do or do not apply to different work cultures.

[5] e.g., http://agilesuperleaders.info/, last checked January 2019.

[6] e.g., https://retrospectivewiki.org/index.php?title=6_Thinking_Hats_Retrospective, last checked January 2019.

[7] e.g., http://tastycupcakes.org/2016/01/the-job-title-game/, last checked January 2019.

**Validity**: In contrast to the team member interview data, the statements of the consultants could not be validated by direct observations or other triangulation.

**Representativeness**: Most of our concepts are grounded in only a small number of instances. Therefore, they are likely to miss a few elements that are in fact common.

**Quantification**: As with any GTM study, the grounding can only provide existence evidence, but cannot measure the relative commonality or importance of different phenomena. A GTM-based study is inherently unsuitable for quantifying phenomena, because no representativity must be assumed. However, it is striking that we found relevant role-clarification dynamics in *all* of our teams and did so without ever looking for them actively –we found these concepts only by deeper analysis of data we had already collected previously. So they appear to be common phenomena in agile teams.

## CONCLUSIONS AND FURTHER WORK

As we have seen, issues with roles in self-organizing teams are diverse and having such issues appears to be common. They can occur at a personal level (finding one's own role) or team level (agreeing on roles). When they are present, they *severely* damage the team's self-organization capability and obstruct the team's work. Ignoring role issues can lead to losing valuable personnel. Agile coaches are often involved (and presumably crucial) in making these issues visible and helping the team resolve them. Resolving roles issues has large practical as well as emotional benefits for the team.

We conclude that our concepts of `Local role clarity` (an individual is aware of a particular role for themselves and accepts it) and `Team-wide role clarity` (each team member is also aware of their team mates' roles, accepts them, and knows how to support them in fulfilling these roles) are worthy of a lot of attention from agile teams. They provide lenses through which teams can address a sizable fraction of their self-organization issues and are a helpful stepping stone for becoming a high-performance team.

Further work should validate the expectation that actively developing a notion of role clarity in a team will lead to improved cooperation.

## ACKNOWLEDGEMENTS

Thanks to Hochschule für Technik und Wirtschaft Berlin (University of Applied Sciences), all teams, experts, and companies who helped.

### Funding

Helena Barke was financed by Hochschule für Technik und Wirtschaft Berlin (University of Applied Sciences). The funders had no role in study design, data collection and analysis, decision to publish, or preparation of the manuscript.

### Grant Disclosures

The following grant information was disclosed by the authors:
Hochschule für Technik und Wirtschaft Berlin.

## Competing Interests

The authors declare there are no competing interests.

## Author Contributions

- Helena Barke conceived and designed the experiments, performed the experiments, analyzed the data, prepared figures and/or tables, authored or reviewed drafts of the paper, approved the final draft.
- Lutz Prechelt conceived and designed the experiments, analyzed the data, authored or reviewed drafts of the paper, approved the final draft.

## Ethics

The following information was supplied relating to ethical approvals (i.e., approving body and any reference numbers):

The need for ethical approval from Zentraler Ethikausschuss (ZEA) of Freie Universität was waived.

## Data Availability

The interview guideline is available as a Supplemental File.

## Supplemental Information

Supplemental information for this article can be found online at http://dx.doi.org/10.7717/peerj-cs.241#supplemental-information.

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
