# Peer review of "Role clarity deficiencies can wreck agile teams"

_PeerJ Computer Science, doi:10.7717/peerj-cs.241_

## Round 0.1 · original submission · Major Revisions

Both reviewers are essentially positive about the paper. The comments are mostly about the presentation. For a qualitative study like this, however, the resulting theory and its presentation are closely intertwined. For that reason I recommend to study the reviewer comments carefully, and follow them up as much as possible to improve the message of this paper.

The key points of the reviewers as I see them include:

1. Improving the overall presentation (R1, point 1, 5)
2. Rethinking how the resulting theory is structured, explaining it in a simple, compelling way (R1, point 2, R2 / "big picture" / "bigger message is lost")
3. Ensuring there is full traceability of the outcomes (R1, point 2, 4)
4. Sanitizing the related work section (R1, point 6, R2)
5. Double checking that the discussion reflects on the results instead of introducing new ones (R2)

Reviewer 1 ·

Basic reporting

There are reporting issues in terms of style, and presentation of the findings. Overall the text is actually very readable, but there are just a large number of issues that require clarification. The presentation of the results needs to be restructured. Diagrams and tables could help to clarify. Details below.

Experimental design

This is a grounded theory study, and more details are needed to explain the analysis. Details below.

Validity of the findings

There seems to be a lot of speculation based on individual/singular observations. The analysis needs to be elaborated, and the findings need to be restructured, which will help in following the argumentation of the authors. Details below.

Additional comments

This article presents a study of self organising agile teams with a specific focus on the dynamics that emerge as team members get to grips with the roles that they have within the team. The study presents a number of findings, for example, that there must be 'local role clarity' (referring to individuals to understand and accept their role) but also team-wide role clarity, meaning that very team member understands and accept their own roles. If these are not present, then agile teams could break down.

This is a very interesting study drawing on a significant amount of data collected, and the authors seem to have very good access to the teams. There is a very nice level of frankness in the findings, and the study could be a great contribution to the agile literature to understand what makes self-organising teams work.

There are, however, a number of issues in the current presentation. I have a list of issues below, and will summarise the key issues here:



1. The paper presentation, while seemingly a trivial detail, is bothering me quite a bit. The use of very short paragraphs and sometimes sections (as short as one or 2 lines) makes little sense in that it raises a lot of questions that remain unanswered. I have listed a number of comments below regarding this issue, but in short, whenever you make some kind of claim or mention some kind of concept, clarify it, don't leave it up to the reader to do the research, but be complete in your presentation. In addition the use of different fonts and colours to distinguish the concepts, and the very formal referencing structures used to refer to individuals (t2dev1) really break the flow of the paper completely.

2. The actual 'theory' or findings should be reorganised and restructured. I find it very hard to clearly pinpoint how to do that, but it seems to me that the various findings are scattered over a number of sections. I suggest that the authors focus on the key findings such as local role clarity and team-wide role clarity, and take these as the central concepts, and link everything else to those. At the moment there are so many observations that seemingly are somewhat unrelated, or at least that relation isn't very explicit. Would a diagram help perhaps?

3. Restructuring of the methodology section; details are scattered over too many subsections, and sometimes related things are not discussed 'together'. (Details below). Further clarity on the analysis is also needed. Suggesting to look at Hoda's studies and notation to identify concepts.

4. Tone down the conclusions that you draw which are not sufficiently supported by the findings. There seems to be a lot of speculation.

5. There are a number of style and presentation issues (besides the font/smallcaps/colour issues in item 1 above). There is a list of issues at the bottom of this review.

6. Related work: the authors state that there is a lot of work on team work, and then simply state that they only discuss a few. I think the whole related work section could be reworked, perhaps some restructuring is needed. I would also strongly suggest to refrain from pointing at any of your findings in the related work section (I don't really like it in Section 1 either in the current form, but I can see why one would do that).


Detailed comments follow below. The review comments suggest I be very constructive and positive - please assume this is implicit. I think this study is very interesting, but have a good few issues, and haven't bothered to tell you how good it all is every time.


Some of the paragraphs are very short, and miss elaboration or explanation. For example, paragraph 3, 4, 5, 6, 7 in Section 1 are very short. It almost reads like a novel. Some of the sections area also very short, e.g. 1.2, 1.3. Why have them as separate sections? Suggest to remove those subsections headers and merge all text under a single header "Section 1". As I proceeded reading the paper, I found that short paragraphs is an issue throughout, some as short as a single line. Please don't do that too much.

Paragraph 3: the present work will characterise one major hurdle; please explain here what the focus is of the paper.

Paragraph 4 is unclear altogether. Not clear what 'intermediate outcomes' are, why abstraction is necessary, and how roles are a form of abstraction. Not sure why Hoda et al 2013 are cited here either.

I'm sure there is a formal definition of role in the management literature. Why not draw on that?

You state that roles are fluent and change over time, which is why you do not name or list specific roles. I can't follow that logic. Also, Scrum defines a number of roles. Why not simply refer to those?

You indicate there is a huge literature on team work from management and other fields, but that only a few are considered. Why? There are likely to be very useful insights in that related work, so skipping it simply because it's from other fields isn't really a valid argument in my view. This is a missed opportunity. Please don't use 'grounded theory' as an excuse.

Sec 2.1 already alludes to some of the findings. I suggest to remove those forward references.

Section 2.1 mentions Tuckman's model, but delays discussion of this to the Discussion section. Why?

Section 2.2 reviews some prior literature organised by author. A better way to conduct a literature review is to synthesise by theme. Instead of Author A found ... Author B found ..., it is better to distill the various themes. Authors may or may not be referenced by name as part of the text. Further, the reference to the authors' own work is too short. This article should stand on its own, so include more detail on that previous work, what you found, and how that led (exactly) to the current study.

Please explain what exactly you don't share in terms of the relevance of personality type. What is wrong with it, in your view? Convince me of your position.

You write that Hoda et al examine roles in agile teams such as Mentor, ... ; actually, those were her findings, she didn't go out to examine those roles; they didn't exist as concepts before her study.

You suggest that Hoda's roles are "fixed" but I don't think she actually claims that her roles are "fixed"; she calls them roles, because that's an easy label that people understand, but it is perfectly acceptable to interpret that as "something that people do" - a "role" that people adopt within the team. I don't think that she suggests that these are fixed roles assigned to people within a team. At least, you don't provide any justification for your claim that she considers them as 'fixed' roles (e.g. quotes from her studies).

Section 2.2 is labeled self-organisation, but section 2.3 also discussed self organisation (e.g. "our contribution is how to define and describe roles in general in self-organised teams"). Consider relabelling your sections or combine them.

Section 2.4 is also too short: just a paragraph. If, as you state, reflection is a central theme in your study, then it warrants more detailed discussion. For example, you use the term reflective practitioner as if the readers are supposed to know what that means, but it is well worth describing what this means, and how this has been considered within the SE literature. A few years ago there was a special issue in IEEE Software for example on the reflective practitioner.


Why did you select Charmazian GT? What were your assumptions and factors in the decision to adopt CGT, as opposed to Glaser or Straussian GT? Simply stating a preference that Charmaz's version of the method has your preference because it's more clear (if that is the case to you) will do. (Glaser is a bit ... vague sometimes, and perhaps you found Strauss & Corbin too detailed?).

Further, you state it is suitable to study agile contexts, citing Hoda and others. Why in particular? You seem to allude to it, but you use signalling words that contrast that (i.e. "Furthermore," as if it is another reason?)

Don't simply refer to Barke and Prechelt when referring to the data used in this study. Again, this paper should stand on its own. Pretend as if your 2018 paper doesn't exist from that perspective, though you probably should cite it as it seems relevant.

When would audio recordings be considered inappropriate? Do you mean inconvenient, or not preferred by the recorded participants? Please clarify.

What do you mean by "These written protocols contain as much verbatim language as possible."? What written protocols? What verbatim language? Is this a good thing? (You lost me there).

You mix the discussion of interviews and observations in the paragraph labeled "Observations". Suggest to not do that.

The paragraph on member check and feedback suggests you completed data collection before analysis; this seems to contradict or conflict the practice of theoretical sampling a bit, because the latter suggests that you keep collecting data to fill in the gaps that emerge during analysis. Please also elaborate what you mean by member check; the term hasn't been used in the main text, only the label.

It is not clear why you state that the expert interviews focused on 'concrete' teams; were the interviews with the teams not regarding 'concrete' teams (namely theirs)?

Suggest to summarise the list of teams in sec 3.3 in table.

Why do you state that you restricted yourselves to Scrum teams 'to limit complexity'? You suggest in parentheses that you did not study other agile methods. But why would that be more 'complex'? Sub-universe is a bit flowery language, in my view.


You mix up the data collection (discussion of teams that you interviewed) and GTM practices (section 3.4); please do that in one place (teams were chosen to cover a variety ...). Further you discuss data analysis (MAXQDA) as part of 3.4 as well.

You separate Data collection (3.2) from 3.4, the actual GT practices which include data collection. I suggest to consolidate that as well.

There should be more detail on the data analysis/coding process. Hoda (and others following her example) has done this well, illustrating with examples how concepts were identified. This helps to understand the process you followed. Further you refer to word-by-word coding but also mention incident by incident coding. Did you use both? More details please.

Section 3.5, first line as a separate paragraph. Following paragraphs are also too short. Please just combine 3.5 with 3.4, or at least present longer paragraphs. The one-liners are not always self-explanatory; what does it mean that "The appearance of a concept name indicates a statement is grounded in the data."? What concept name; what do you mean by appearance, even? I can't follow this.

You write "a quote is often (if not always) provided ..." But you know whether it's often or always, because you wrote this paper. Be more precise.

I don't understand the reference / metaphor in footnote 3 on page 6 to the Facade layer pattern. Suggest to simply remove.

Sec 4.1, 2nd para, "This narrative" - which narrative? "This" is now a dangling pointer, because it's a separate paragraph - that's why paragraphs should not be split up!

Sec 4.1, "So here is the set of concepts ..."; this isn't great style, to be honest. Don't just list the concepts and announce them as "here is the list of results": tell a story instead.

I can't follow what the finding is in Sec 4.1 You present S1, S2, as definitions, perhaps?
I suggest to explain and describe these concepts, indicating what they refer to and illustrate them with events/quotes/other indicators/evidence that led you to the coining of these concepts. I also strongly suggest to remove the numbers/prefixes S1.2-, S1.1-, as it breaks the flow.

In sec 2.3 you suggested that you did not consider personality type research, but in the first quote of Sec 4.2 you refer to this very concept: "he has a different personality". This is confusing. Why do you not consider personality relevant then, but still use it in a quote to illustrate your findings?

Section 1 introduced the key findings as F1, F2, etc. These labels are also used in the findings section, but I think these labels should not be used in the presentation of the results, but could only be used when summarising the results. In other words, remove any reference to Fn, but theorise instead in the presentation of the result. The presentation of the key findings can then be done in a summarising section.

Some quotes are simply not very illustrative. Why did you select the quote on "meditation", for example? What does this illustrate?

I cannot follow why the quote by t2dev1, page 7, second last quote, is labeled "effective feedback". I think it's pretty bad, to state "therefore I cannot respect you as a developer". I'm surprised nobody quit yet.

Please refrain from speculation, e.g. page 8: Now that t2dev2's role is clear, knowledge transfer from t2dev1 will likely become a lot faster and the overall team capability may become better more quickly. You conclude very firmly that "the problem is a local role clarity" - but that's just your theory, really - a hypothesised mechanism at work. What evidence do you have for this at all?

A main problem I have while reading the results is that there is a lot of stuff going on, but it's really hard to maintain a clear (clean) record in my head of what it is you actually have learned/found through this study. No doubt that this is partly due to the presentation style (small caps, different colours), the use of too formal systematisation of references to individuals (t2dev1), and the very short paragraphs and section sometimes which break the flow and concentration, but there is a different issue. I appreciate the difficulty of writing grounded theory, but there seem to be so many findings that are not clearly discussed in separate sections. Some sections eg 4.4 are too short. One sentence and one quote, that's not enough to convince me of your argument that, 'once role clarity is achieved, technical expertise becomes less important'. That's a single observation, but there is no additional evidence presented for that. The labelling of that section (4.4) is ("having non-perfect technical expertise") also not very descriptive.



Language & Style

we need to abstract a bit --> rephrase "a bit".

I'm not convinced that typesetting the concepts identified through GT in small caps is a useful practice; it distracts from reading. When typesetting "Roles" for example, "Role" is in small caps/red, and the s is black in normal font; this also doesn't help. References to other roles like sm, po are also hard to interpret, breaking the flow of reading. Strongly suggest to just spell it out in normal font.

Amount of meetings --> Number of meetings. Use "amount" to refer to inseparable things like water and beer. Use Number to refer to distinguishable, concrete items.

The term "data" is plural.

Suggest to remove unnecessary words, e.g. "In total, the present research" --> This research. Simpler is better.

Sound recordings: probably audio recordings. Sound is what you hear in the disco, hopefully, and preferably not in an interview. You wouldn't be able to understand people.

The referencing of teams as t1, t2, etc.: I suggest to simply state Team 1. Further I think the mathematical labelling of t4dev2 etc. to refer to team 4, developer 2 is a bit much. Does it matter for the reader whether it was dev2 or dev3?

4.2, Comparatively obvious - do you mean relatively obvious? Even that is a bit vague. (Comparatively - with what?)

The use of "I" in Sec 4.2, you writing as a researcher to "quote" participants (?) is a bit vague. The changing of "I" and "we" (as authors) throughout the paper is confusing and should be removed.

Reduce informal language such a "a bit" (also listed above) and "So the findings are", "the point is" etc. Please also try to remove "glue" language like "now we will explain". Don't announce it mid-text, just do it.

"Indeed," cannot be used at the start of a new paragraph. Indeed, (no pun), the word is often overused (in general, not necessarily this paper), and is a signalling word that highlights a previously stated argument. (Top of page 8)

Please refrain from using "variables" in the text, such as "someone else's role R.", so hat you can refer to "R" as someone else's role later. This is not readable. (Same for "team member A").

·

Basic reporting

This paper presents a theory on the role of clarity deficiencies to help the workings of agile teams. Using a Grounded Theory Methodology on five agile team the study concludes that lack of role clarity, both at an individual and team level, creates friction in team, hampers everyday work and can also lead to employee turnover. The study concludes a high need for role clarity by whatever means it takes.

Overall, the paper is easy to understand with great attention paid to details. There are some places though where additional information can help.

Related work section feels more like a mix of background and related work. In this case, it might be worth presenting it the same way. In related work section, the relevance of each sub-section isn’t quite clear. Subsection ‘Teamwork in general’ actually doesn’t talk about teamwork in general but about specific studies which lead us to current work. At some point why these subsections exist at this place is not clear. For instance, what is the purpose of subsection ‘reflection’? What is the bigger message that is intended to be conveyed here?

Similar issues exist in the results section. Each subsection presents a distinct aspect of the observation, but it is not evident what the big picture is. It might help to present an overview of the factors that play a role in getting role clarity before delving into details. For instance, subsection ‘having non-perfect technical expertise’ is actually a positive use-case of role clarity. I suggest revisiting the title of each subsection to better convey the story.

At some places, concepts from related studies are poured in without explanation. For example, Constructing Grounded Theory variant of Charmaz and MAXQDA. It will help the reader if a brief explanation is provided for the terms and concepts borrowed from the literature.

Experimental design

The objective to help teams self-organize in an agile set-up is a relevant problem and building a theory around it seems appropriate. The experimental design to construct theory is presented in great details and all necessary precautions seem to have been taken. There are a few details that raise doubt.

Section 3.2 reports that "small talks built trust and lowered the hurdle to...". How are such things substantiated? Was this a part of the plan?

I also miss details at places on reaching conclusion. For instance, in section 4.1 it is reported that “it appears that TEAM WIDE ROLE CLARITY is as important for a team as is a sufficient level of technical expertise.” Details on how this statement is reached and similar statements will greatly help in appreciating the results. In its current form, not all claims seem to be backed by evidence.

Validity of the findings

The paper builds a theory based on discussions and observations of teams. To what extent do these results hold? Do all teams agree to the theory? Do they have an alternate explanation?

Discussion section introduces new findings not presented in the results section. For example, section ‘How to do it?’. I am not sure how these results are derived or how generalizable these results are? It will help to either make these questions a part of study design or refrain from describing it.

The paper reports that no tangible positive outcomes were seen when the concepts were applied in practice. What potentially explains it? Does that mean that the proposed framework is of no use?

Additional comments

The paper aims to present a theory for helping teams self-organize. But it seems that the bigger message is lost in the great details presented in the paper. The paper talks about too many things but fails to organize it in a way that helps reader get a bigger and coherent message.

---

## Round 0.2 · Minor Revisions

It is good to see that the authors have thoroughly revised the paper, taking the reviewer's feedback into account.

The paper introduces important concepts surrounding role clarity in agile teams, distinguishing local from team wide role clarity. The paper employs grounded theory to develop conceptual concepts and relationships from 16 interviews with experts, as well as 100 pages of observations from team interactions. Role clarity involves role acceptance, role expertise, and role execution, as well as team and self reflection. The paper argues that role clarity helps teams in handling (lack of) expertise and dysfunctional behavior, and that role vagueness hampers self organization.

Reviewer 1 has a number of suggestions that will help to drive home the core message of this paper in a better way. I therefore recommend acceptance with a minor revision, in which the authors get the opportunity to take reviewer 1 suggestions into account.

Reviewer 1 ·

Basic reporting

Please general comments; sometimes too much "casual language".

Experimental design

N/A.

Validity of the findings

No issues.

Additional comments

This revision is a big improvement on the initial submission. I think this paper present some interesting findings, and would support publication - I think it is converging to an acceptable state. I have a number of small issues, suggesting a minor revision.

In sum, my comments refer to style and presentation
1. Style: the paper still exhibits some casual use of language, or language that one would use in daily conversation rather than what one would expect from an academic paper.
2. Presentation: the paper still has some sections that are too heavy on bullet points, suggesting this is a slide deck; I think this really needs to be polished into proper text rather than trying to summarise everything in bullet points.


Abstract

"They should invest whatever it takes"

The "whatever it takes" phrase might be too open-ended...
Perhaps something more neutral (and less dramatic) might be more appropriate academic style.


Introduction
- why do you need "(supposedly)" ? Can it be removed?
- misplaced closing parenthesis in l.55
- paragraph l.56-59: short sentences that don't really flow. "There is research aboutself-organized teams in general." - this is a bland statement - link it to your study here.
- l. 60 since a few years, *we have worked* (not: we work)
- l.61 no need for "own" - you already state "our"
- If you state "in our first article", the reader might expect that you have a 2nd aritle (is that this one); it seems to signal something as earlier you stated "in our previous work". But "previous work" and "first article" are the same I think?
- Still think that subsections 1.1 and 1.2 don't need a subsection title - in other words, suggest to remove those headers (but keep the text). They seem tagged on to the main part of Sec 1.



Method
- picked -> selected
- happen to have different levels of maturity -> seems a "coincidence" and lack of deterministic research design. Suggest to remove "happen to".
- l.94 "will be required" -> suggest to remove future tense such as this throughout the paper; there's no need for it; instead, starting after the comma on l.94: ", which requires a thorough analysis so as to reveal recurring structures across teams."
- l. 95: "some longitudinal data" -> be more specific; this sounds a bit ad hoc and hesitant ("some data").
- suggest to replace Helena Barke with "the first author".
- l.101 -which we will call -> which we refer to as (again, no future tense); same sentence: period missing.

- Data collection WAS done. Missing period after Table 1^3
- this research -> this study

Sec 2.3
- we will now explain -> we now explain (no future tense)
- l.161 we will ... ->remove will. (I won't comment on this issue again, please check the whole paper for this)

l.169 - what do you mean by "phenomena"; why do you need to make explicit that you do not focus the discussion "around people or teams"?

Section 3
- what are the labels R1-Accept, R2-Expertise, and R3-Acting again?
Why you need these labels? Are these attributes of a "role"? Make this more clear.

- I'm visually oriented, but am aware not everybody is. Does it make sense to add a diagram that captures the overall results as presented in Sec 3.1? Concepts can be boxes, theorised links as arrows?

- What does "No. 2" and "No. 3" mean (l. 216 and later)

- language such as "The point is" (l. 252) sounds like daily, verbal language rather than written language that belongs in an academic paper. Suggest to rephrase. The use of "So" at line 347 ditto. The style has certainly improved w.r.t the first submission, but on these occasions the authors still write as one would talk in a conversation.

Section 4
Suggest to elaborate briefly on each of the 7 "findings" in 4.1
Don't link them together as in the case of 3 and 4 with the word "but"


4.2:
l. 433 "In the following Section"
- do you mean THIS section?
- don't capitalise Section if you just use the word "section"w without referring to a specific numbered section (cf. "in this section" vs. "in Section 3")
- as mentioned, no future tense (no "will")

I'm usually not a fan of delaying a presentation of related work until so late in the paper, but recognise cases where this can be useful; in such cases it needs to be contrasted very clearly with the current study's findings. Is it worth adding a table that juxtaposes related work with this study, if you do decide to keep the related work until after the presentation of the study instead of as section 2.

It seems to me that Section 4.3 would be better labeled as Implications for Practice. Again, a table might be useful that juxtaposes the specific findings and implications that result from each of them. Given your findings what should one now do in practice (and what might the implications for research be? Does your research open up new questions?)


Section 5
Strongly suggest to rid of the bullet points; make this normal text.
Don't use the term 'proof'; you provide evidence not proof.

Section 6
Don't use an enumeration of findings or conclusions. Write text. This is not a powerpoint presentation published in PeerJ. This is an academic paper.

Personnell is spelled as personnel

·

Basic reporting

Paper is well articulated and precise. The message of the paper comes out quite well.

Experimental design

Paper is clear about the method and scope of the results is well defined.

Validity of the findings

Results seem sound and not overclaiming.

Additional comments

This version of the paper brings the core message clearly and with scientific accuracy. The outcome of this study is a useful contribution towards building a theory on the dynamics in self-organizing teams.

My only suggestion is correcting the typing errors in the paper.

---

## Round 0.3 · accepted · Accept

Thank you for your careful update of the paper, addressing the main reviewer feedback. The paper has an original take on agile team work. It has a clear and simple message: get your roles clear. The paper is relevant to both research (in terms of methods, findings, and future directions) and practice (with clear implications for current teams). It is my great pleasure to recommend acceptance of this paper.